ecology/evolution

Chlorophyceae, disruptive selection, gamete size, meiotic progeny, Parker, Baker and Smith's model, Ulvophyceae

**Author for correspondence:**
Tatsuya Togashi
e-mail: togashi@faculty.chiba-u.jp

# A comparative test of the gamete dynamics theory for the evolution of anisogamy in Bryopsidales green algae

Tatsuya Togashi[1], Yusuke Horinouchi[1] and Geoff A. Parker[2]

[1]Marine Biosystems Research Center, Chiba University, Kamogawa 299-5502, Japan
[2]Department of Evolution, Ecology and Behaviour, University of Liverpool, Liverpool L69 7ZB, UK

TT, 0000-0002-9270-9863; GAP, 0000-0003-4795-6352

Gamete dynamics theory proposes that anisogamy arises by disruptive selection for gamete numbers versus gamete size and predicts that female/male gamete size (anisogamy ratio) increases with adult size and complexity. Evidence has been that in volvocine green algae, the anisogamy ratio correlates positively with haploid colony size. However, green algae show notable exceptions. We focus on Bryopsidales green algae. While some taxa have a diplontic life cycle in which a diploid adult (=fully grown) stage arises directly from the zygote, many taxa have a haplodiplontic life cycle in which haploid adults develop indirectly: the zygote first develops into a diploid adult (sporophyte) which later undergoes meiosis and releases zoospores, each growing into a haploid adult gametophyte. Our comparative analyses suggest that, as theory predicts: (i) male gametes are minimized, (ii) female gamete sizes vary, probably optimized by number versus survival as zygotes, and (iii) the anisogamy ratio correlates positively with diploid (but not haploid) stage complexity. However, there was no correlation between the anisogamy ratio and diploid adult stage size. Increased environmental severity (water depth) appears to drive increased diploid adult stage complexity and anisogamy ratio: gamete dynamics theory correctly predicts that anisogamy evolves with the (diploid) stage directly provisioned by the zygote.

## 1. Introduction

In eukaryote sexual reproduction, a dominant explanation for the evolution of anisogamy (the production of gametes of different

sizes between males and females) from ancestral isogamy (the production of gametes of equal size between mating types) is Parker, Baker and Smith's model (hereafter, the PBS model) [1]. It relies on gamete competition and has been highly modified from the original [2–6]; PBS merges with gamete limitation into one general model (i.e. gamete dynamics model or GD model) [4,6]. Anisogamy causes morphological and behavioural differences between males and females through sexual selection [1,7–10], and its evolution, therefore, represents one of the most fundamental problems in evolutionary biology [6]. GD explains disruptive selection on gamete size based on two opposing selection forces: smaller gametes are favoured because more can be produced, and larger gametes are favoured because they provide the zygote with additional resources that increase its survival and/or other aspects of fitness. GD predicts increased selection for anisogamy as zygote size increases.

The GD model assumes that (i) zygote fitness more than proportionally increases with its size (volume), at least over part of the zygote size range [1,3], and that (ii) a size-number trade-off applies, i.e. gamete size decreases in inverse proportion to the number of gametes produced from a given total gametic investment, a trait confirmed empirically in many organisms including algae [11]. The evolutionarily stable strategy (ESS) changes from isogamy to anisogamy as the importance of zygote size for fitness increases [3,4]. Zygote size is predicted to correlate positively with adult size and complexity because more resources could be useful for large complex organisms to develop [1,3,12], and the anisogamy ratio (female/male gamete size) is expected to increase with adult size and complexity [1,3,12].

Our aim here is to test these GD predictions using the Bryopsidales, a green algal order in the class Ulvophyceae, though the traditional testing ground for GD has been volvocine green algae with haplontic life cycles that have a haploid adult (gametophyte) stage without a diploid adult (sporophyte) stage (figure 1a) [14]. Here, we use the term 'adult' for both the fully grown (diploid) sporophyte and fully grown (haploid) gametophyte. Bryopsidales comprise mostly macroscopic siphonous marine green algae, and many species are dioecious [14], although the evolution of anisogamy can also be modelled in hermaphrodites [2,15–17]. Some taxa have diplontic life cycles without a haploid adult (gametophyte) stage (electronic supplementary material, table S1), as do higher plants and most animals: after syngamy, the zygote develops directly into the diploid adult (sporophyte) in which meiosis occurs during gametogenesis [14], and the gametes are the only haploid stage (figure 1b). However, other taxa have haplodiplontic life cycles (electronic supplementary material, table S1) in which the diploid zygote grows into a diploid adult (sporophyte) before undergoing meiosis, mitosis and release of many unicellular haploid progeny (stephanokont zoospores) [18,19] (figure 1c). Each zoospore develops into a haploid adult (gametophyte), which may be multicellular and complex, and which releases gametes to produce the following generation. There is no obvious translation of resources from the diploid zygote to the haploid adult—it is, therefore, unlikely that anisogamy can correlate with the need for increased zygote resources to facilitate the development of the large complex haploid adult. This leads us to propose and test a refined prediction of the GD model that the anisogamy ratio may increase with the size and complexity of the diploid, rather than the haploid adult stage, because only the diploid stage is directly provisioned by the zygote.

The species in this order exhibit useful traits for comparative tests of GD theory. First, various anisogamy ratios are observed (electronic supplementary material, table S1). Note that each sporophyte in diplontic species (figure 1b) or gametophyte in haplodiplontic species (figure 1c) usually consists of a single immense multinucleated cell with a wide size range [14], which allows large gametes to be produced [5]. Therefore, cellularity is not related to body size [20]. In Bryopsidales, as the original GD model assumed [1], individual gametes of both sexes are separately and synchronously released into a medium without parental care of zygotes [21]; the exception is the genus *Dichotomosiphon*, which differs from all other Bryopsidales by inhabiting freshwater, exhibits oogamy with sperm and eggs, and females retain their eggs (electronic supplementary material, table S1).

The gametes of ulvophycean green algae use phototaxis and sexual pheromones to increase fusion efficiency. Many isogamous and slightly anisogamous species that inhabit the shallowest water have tidal-linked mechanisms to release gametes synchronously [22,23]. Gametes from both sexes have an eyespot and show positive phototaxis [24] (electronic supplementary material, figure S1a,b); they gather under the two-dimensional surface of the sea and fertilize effectively [24]. Phototaxis becomes negative immediately after sexual fusion [24]. Markedly anisogamous species, including Bryopsidales species, often inhabit relatively deeper waters and use light level to synchronize mass-spawning behaviour in the morning on a given day [25–27], with different species spawning at different times [25]. In such species, male gametes lack an eyespot (electronic supplementary material, figure S1c), but female gametes have

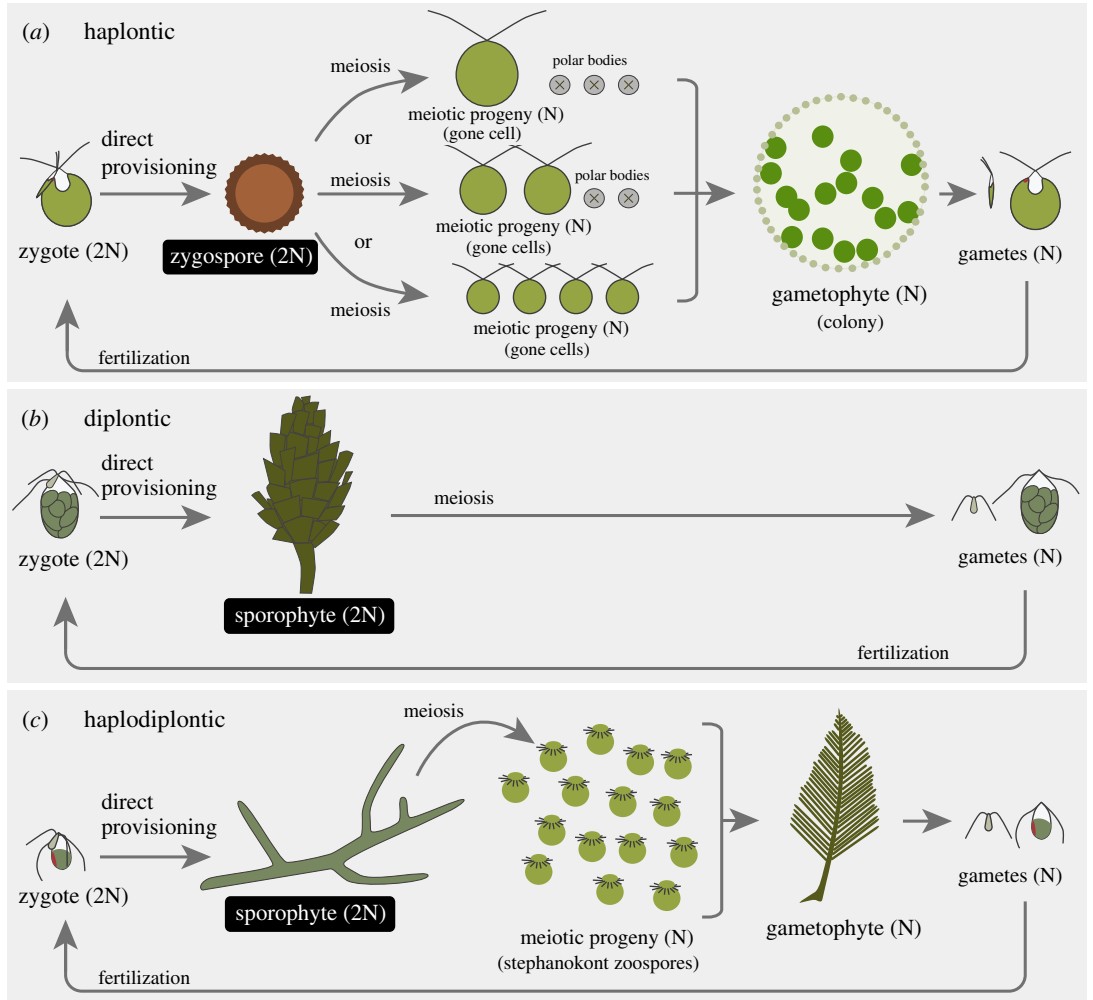

**Figure 1.** Life cycles and the production of meiotic progeny in green algae. N: haploid; 2N: diploid. (*a*) Volvocine green algae with a haplontic life cycle [10]. Each zygote produces a diploid zygospore (i.e. a zygote directly provisions the diploid zygospore). One, two or four meiotic progeny (gone cell(s)) are produced by a zygospore through meiosis. When a lower number of meiotic progeny (one or two) are produced, the remaining meiotic products are discarded as polar bodies. Each gone cell produces a haploid gametophyte. (*b*) Bryopsidales green algae with a diplontic life cycle. Each zygote produces a diploid sporophyte (i.e. a zygote directly provisions the diploid sporophyte), and meiosis occurs during gametogenesis in the sporophyte. (*c*) Bryopsidales green algae with a haplodiplontic life cycle [13]. Each zygote produces a diploid sporophyte (i.e. a zygote directly provisions the diploid sporophyte). Numerous meiotic progeny (stephanokont zoospores) are produced by a diploid sporophyte, and all resources are used simultaneously (holocarpic). Each zoospore develops into a haploid gametophyte.

one [27] (electronic supplementary material, figure S1*d*). Non-phototactic male gametes are attracted by sex pheromones to the positively phototactic female gametes [27]. Species that produce non-phototactic gametes without an eyespot in both sexes (electronic supplementary material, figure S1*e,f*) are found in the deepest water [28]. A possible interaction between the evolution of anisogamy and motility dimorphism has been analysed theoretically [29]. The gametic systems (isogamy, anisogamy) of ulvophycean green algae seem to be linked to habitat [30]. However, few analyses of gamete size and habitat have been performed. Therefore, we explore an alternative evolutionary driver of increased anisogamy ratio: zygote size increases with environmental severity in habitats [31], where additional zygotic resources increase survival [5].

We also wish to investigate the GD theory prediction that in cases where strong anisogamy evolves, male gametes will be minimized by gamete competition or limitation, independently of environmental forces shaping zygote size [5]. A minimum viable gamete size [3,12,32] is often invoked because without a minimum limit, depending on the relationship between zygote size and fitness, isogamy with gametes of hypothetical size zero can be the ESS [33]. Whether male gametes have been minimized should be open to empirical testing [33].

We analysed comparative data on gamete size and behaviour, and the size and complexity of the diploid adult (sporophyte) stage using phylogenetically controlled methods, as follows.

(1) We sought evidence that male gametes have been selected towards some minimum optimal size related only to effective fertilization during spawning. Though all male gamete sizes may not have strictly equal values, even among closely related species, we predict that male gamete size will be: (i) for all species smaller than the female gametes of any species; (ii) less variable than female gamete size: the coefficient of variation (the ratio of the standard deviation to the mean) for male gametes will be less than that for female gametes; and (iii) unrelated to female gamete size, which will be varyingly optimized by number produced versus survival as zygotes. If male gametes are minimized, both zygote size and the anisogamy ratio will depend on female gamete size.

(2) We compared female gamete sizes in species with and without an eyespot. Because inhabiting greater depths imposes restriction in light and temperature that probably demand greater zygote resources, and because the sea surface might be too far for gametes to reach by positive phototaxis, we expect zygote size to correlate positively with their sea depth of habitat, and female gametes without an eyespot to be larger than those with an eyespot.

(3) We tested our hypothesis that the anisogamy ratio correlates positively with the size and complexity of the diploid adult (sporophyte) rather than that of the haploid adult (gametophyte) stage.

Our results support these GD theory-based predictions: (i) male gametes are minimized, showing little size variation; (ii) female gametes show considerable size variation and those without an eyespot are larger than those with an eyespot; and (iii) the anisogamy ratio correlates with diploid adult (sporophyte) stage complexity and indicates the critical roles of the environment (as a driver of zygote size) and the diploid stage in the evolution of anisogamy in Bryopsidales green algae.

# 2. Methods

## 2.1. Life-history data of Bryopsidales

Most data on gamete size (i.e. the length of the major and minor axes of gametes of each sex), the presence of an eyespot or phototaxis in each sex, and adult gametophyte sizes of 32 taxa of Bryopsidales representing eight families were obtained by literature searches (electronic supplementary material, table S1). For *Bryopsis maxima*, gamete size was measured from published micrographs with its scale. For *Bryopsis corticulans*, we obtained gametes of both sexes by culturing male and female gametophytes in our laboratory and measured the size of gametes for each sex directly (see [34] for the culture methods). We calculated the volume of gametes by assuming that each gamete was ellipsoidal in shape, as has been confirmed empirically [11]: $V = \pi ab^2/6$, where $V$, $a$ and $b$ are respectively the volume, the length of the major axis and the length of the minor axis of the gamete. The anisogamy ratio ($\alpha$) is defined as follows: $\alpha = V_f/V_m$, where $V_f$ and $V_m$ are the volumes of female and male gametes, respectively [35]. We took the cube of the maximum length of the final-size sporophytes to represent the relative volumetric size of diploid adults in each species, because Bryopsidales sporophytes are often filamentous, and their diameter appears to increase in roughly constant proportion to their maximum length, so that maximum length cubed gives a measure of relative volume.

## 2.2. Phylogenetic relationships of Bryopsidales

We used the plastid-encoded RUBISCO large subunit (*rbc*L) sequence-based maximum-likelihood (ML) phylogeny originally constructed by Lam & Zechman [36] and calculated branch lengths (see [36] for detailed methods). This phylogeny contains the above 32 Bryopsidales taxa for all phylogenetically controlled analyses conducted below. ML is a robust method [37] that produces quantitatively and qualitatively similar results to the weighted neighbour-joining method of tree construction [38]. There was no significant difference between the ML tree and the all-else-equal most parsimonious trees (Shimodaira–Hasegawa (SH) test [39], $0.212 < p < 0.325$) [36]. The ML tree was also not significantly different from the Bayesian inference consensus tree (SH test, $p = 0.483$) [36]. The topology of Lam & Zechman's phylogeny [36] has been mostly maintained in other phylogenetic relationships in siphonous green algae (Bryopsidales and Dasycladales) analysed by Verbruggen *et al.* based on a five-locus data matrix, in which relaxed molecular clock methods calibrated with fossil records was used [40]. More taxa for which we could obtain life-history data are included in Lam & Zechman's phylogeny [36] than Verbruggen *et al.*'s phylogeny [40].

## 2.3. Comparative tests of Parker, Baker and Smith's theory for anisogamy in Bryopsidales

First, we compared the coefficient of variation (CV) of gamete volume between males and females and analysed the relationship between male and female gamete volume using the phylogenetic generalized least-squares (PGLS) method [41]. Second, we compared the volume of female gametes between taxa with and without an eyespot using PGLS ANOVA [42]. (Notably, PGLS generally analyses the linear relationship between an explanatory variable and a response variable. Therefore, we did not set the male ($V_m$) and female ($V_f$) gamete sizes or zygote size ($V_m + V_f$) and the anisogamy ratio ($V_f/V_m$) as explanatory variables and the response variable, respectively, as the response variable included the explanatory variable. This decision was made because the relationship between an explanatory variable and a response variable cannot be described by a first-degree equation.) Finally, we examined the relationship between the volume and complexity of the diploid adults (sporophytes) and the anisogamy ratio. To evaluate the volume of the diploid adults (sporophytes), we scored 0 (less than 1 cm³), 1 (1–10³ cm³), 2 (10³–10⁶ cm³) and 3 (greater than 10⁶ cm³). We gathered data on five discrete complexity traits in the diploid adults (sporophytes): (i) multinuclearity, (ii) interwoven siphon, (iii) segmented body, (iv) stipe and capitulum, and (v) three-dimensional body design. For each species, each trait was scored 1 (presence) or 0 (absence), then summed to give an overall complexity score. We used PGLS to regress the volume and the number of complex traits with the anisogamy ratio. The comparison of the CV was made by the R package cvequality [43]. PGLS and PGLS ANOVA were implemented in the R package caper [44]. All statistical analyses were performed in R (v. 3.2.3) [45].

# 3. Results

## 3.1. Gamete size (volume)

All species recorded were anisogamous, with the male gamete defined as the smaller gamete. The relationship between male and female gamete volumes is shown in figure 2a. Except for *Bryopsis hypnoides* and *Codium fragile*, in which the male gametes were slightly larger than the female gametes of the weakly anisogamous *Caulerpa brachypus*, all male gametes were smaller than any of the female gametes across species (see also electronic supplementary material, table S1). In these species, the CV of the male gamete volume ($CV_{male} = 0.34$) was significantly much smaller (less than 11%) than that of the female gamete volume ($CV_{female} = 3.21$) ($p = 0.01$; electronic supplementary material, figure S2), and there was no correlation between the male gamete volume and the female gamete volume (PGLS, $p = 0.26$, adjusted $R^2 = 0.03$).

## 3.2. Comparison of size (volume) between female gametes with and without an eyespot

The presence of an eyespot in male and female gametes is shown in the electronic supplementary material, table S1. In all taxa for which we had data, the male gametes did not have an eyespot. By contrast, the female gametes in some taxa had an eyespot, though the female gametes of other taxa did not. The female gametes without an eyespot were significantly larger than those with an eyespot (PGLS ANOVA, $F_{1,12} = 9.17$, $p = 0.01$; figure 2b). Environmental habitats and anisogamy ratio that the size and phototactic behaviours of gametes (figure 2a,b) suggest are shown in figure 2c.

## 3.3. Relationships between the size (volume) and complexity of the diploid adults (sporophytes) and the anisogamy ratio

First, we found no significant correlation between the volume of the diploid adults (sporophytes) and the anisogamy ratio (PGLS, $F_{1,10} = 0.0123$, $p = 0.913$, adjusted $R^2 = -0.09$; figure 3a; see also electronic supplementary material, tables S1 and S2). Second, the relationship between diploid adult (sporophyte) complexity and anisogamy ratio is shown in figure 3b (see also electronic supplementary material, tables S1 and S2). Analyses included the 16 terminal taxa with available data for all five traits. Species with a greater anisogamy ratio have significantly more complex traits (PGLS, $F_{1,10} = 7.38$, $p = 0.02$, adjusted $R^2 = 0.37$). Notably, the relationships between the size (volume) and complexity of the diploid adults (sporophytes) and the size (volume) of zygotes were similar to the relationships between the size (volume) and complexity of the diploid adults (sporophytes) and the anisogamy ratio above: there was no significant correlation between the volume of the zygotes and the diploid

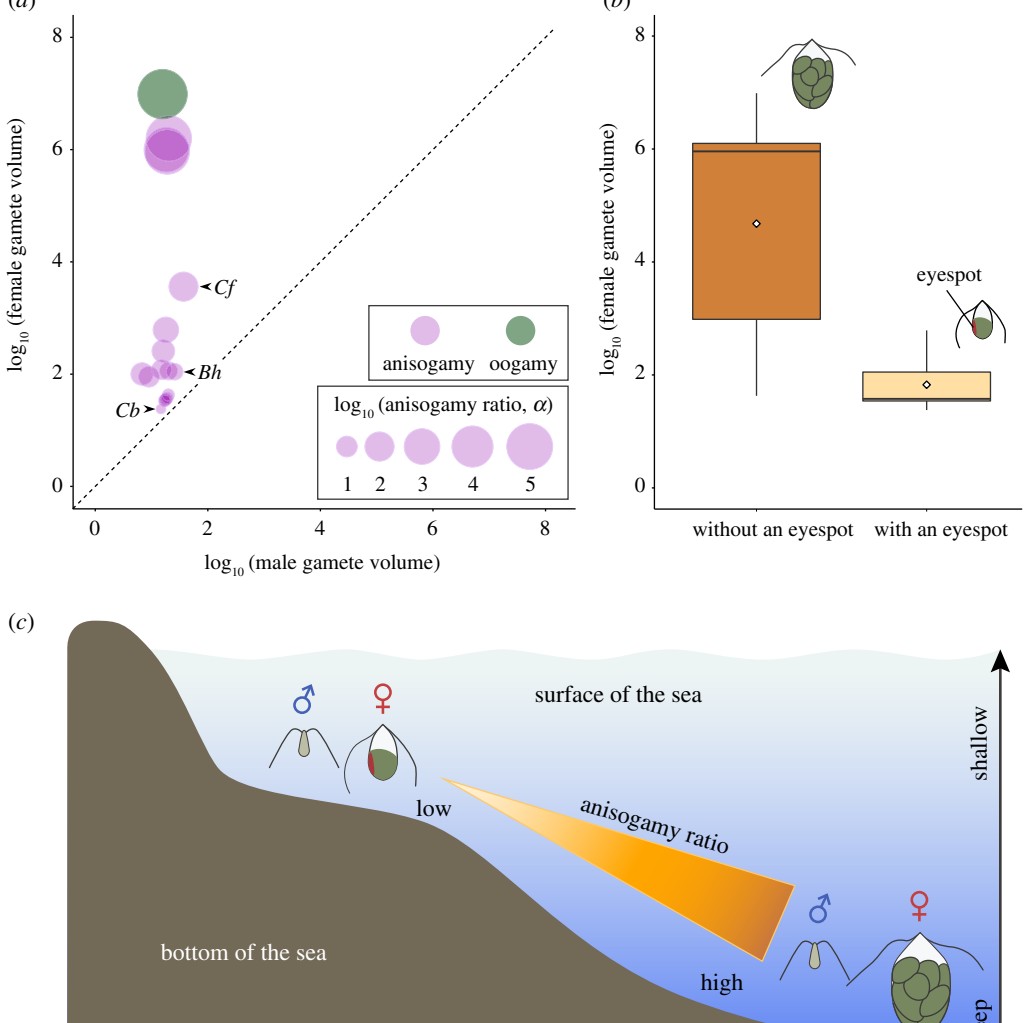

**Figure 2.** Analyses of Bryopsidales comparative data on gamete size and environmental habitats. See Methods section for the definition of the anisogamy ratio, $\alpha$. (a) Relationship between the size of male and female gametes. Arrowheads indicate *Bryopsis hypnoides* (*Bh*), *Codium fragile* (*Cf*) and *Caulerpa brachypus* (*Cb*) (see text for details). The dotted line indicates isogamy ($\alpha = 1$). (b,c) Gamete size and environmental habitats. (b) Sizes of female gametes with and without an eyespot. Boxes represent the interquartile range (IQR) between the first and third quartiles and the line inside represents the median. Whiskers define the lowest and highest values within 1.5× the IQR from the first and third quartiles, respectively. Diamonds indicate means. (c) Environmental habitats and anisogamy ratio that the size and phototactic behaviours of gametes (a,b) suggest.

adults (sporophytes) (PGLS, $F_{1,10} = 0.21$, $p = 0.67$, adjusted $R^2 = -0.08$; electronic supplementary material, figure S3a and tables S1 and S2); diploid adults (sporophytes) that develop from larger zygotes have significantly more complex traits (PGLS, $F_{1,10} = 6.47$, $p = 0.029$, adjusted $R^2 = 0.33$; electronic supplementary material, figure S3b and tables S1 and S2).

# 4. Discussion

## 4.1. Gamete size evolution in Bryopsidales

Our results indicate that male gametes have been minimized in most species of Bryopsidales, tending towards a critical minimum size below which further reduction does not occur (figure 2a). Across species they are almost always smaller and much less variable in size than any female gametes (in contrast to volvocines [46]). Further, they show organelle reduction. Male gametes have lost eyespots and have transparent degenerated chloroplasts [14] (see also electronic supplementary

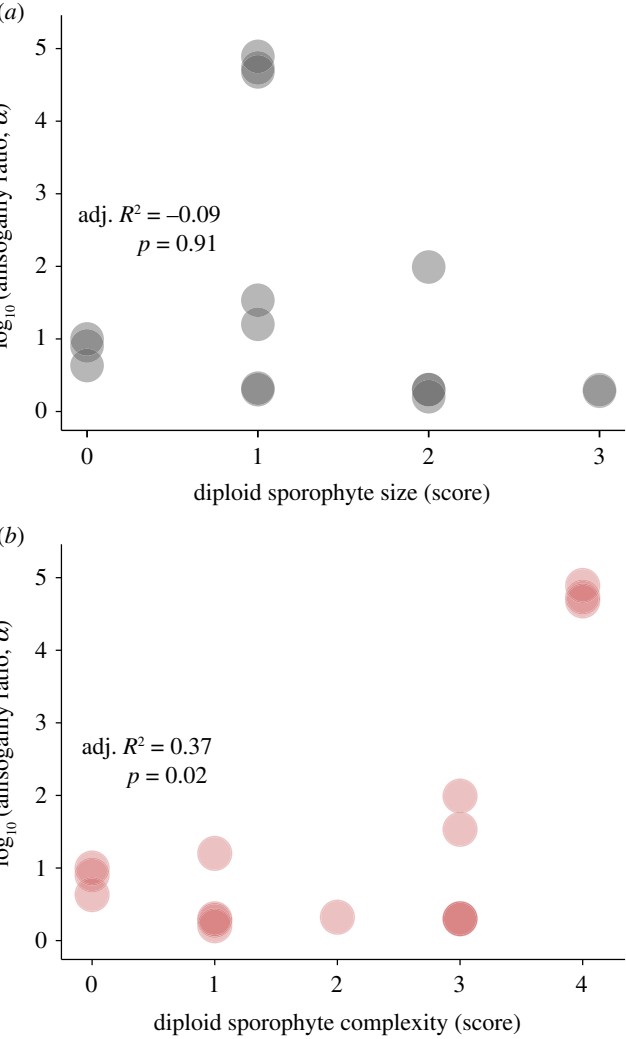

**Figure 3.** Analyses of Bryopsidales comparative data on size and complexity in the diploid adult sporophyte stage with the anisogamy ratio (see Methods section for the definition of the anisogamy ratio, $\alpha$; see also electronic supplementary material, tables S1 and S2). (a) Relationship between diploid adult sporophyte size and anisogamy ratio. (b) Relationship between the complexity of the diploid adult sporophytes and the anisogamy ratio.

material, figure S1c,e); digestion of male chloroplast DNA occurs during male gametogenesis and mitochondrial DNA also disappears in almost all male gametes [47]. Chloroplasts and mitochondria are uniparentally inherited only from female gametes [47]. These results all comply with basic predictions of GD theory, namely that selection for maximum productivity drives male gametes towards the minimum viable gamete size, while female gametes evolve to a larger size, varying with the species-specific demands of zygote provisioning. In contrast to the female gamete and hence the zygote, the male gamete has primarily a transient adaptive function—that of fusion. We suggest that this transient (and relatively invariant) function is the main reason why male gamete minimization has occurred, in conformity with GD theory. Because of male gamete minimization, both zygote size and anisogamy ratio correlate positively with increasing female gamete size.

We suggest that different selective forces have affected the phototactic behaviour of gametes of the two sexes. In all habitats, male gametes lack an eyespot and show no phototaxis in all taxa for which we had data (electronic supplementary material, table S1), possibly representing organelle reduction owing to male gamete minimization. Sexual pheromones released by female gametes might also have played an important role in permitting the evolution of eyeless male gametes [27]. Note that ulvophyceans with phototactic gametes and/or sexual pheromones might not be often under gamete limiting conditions, so that gamete competition is more likely to account for our results. Species in which gametes of both sexes lack eyespots often inhabit deep water [28], and female gametes in species lacking eyespots (electronic supplementary material, table S1) are larger than those with an

eyespot (figure 2*b*). They typically contain more chloroplasts [14] (see also electronic supplementary material, figure S1*f*). Their increased size may reflect an adaption to severe environments in deep water where photosynthesis is highly constrained [5]: light intensity is low, and the water is cold, and eyespot loss may have resulted from the difficulty of reaching the sea surface using positive phototaxis (i.e. environmental restriction) (figure 2*c*). Losing the eyespot may permit energy to be diverted into chloroplasts. Our results suggest that in Bryopsidales green algae, anisogamy evolved with habitat: deep water habitats generated selection to increase zygote size as most small zygotes would not be able to survive there even if more zygotes were produced hence increasing the anisogamy ratio [3].

## 4.2. Life cycles and the evolution of anisogamy in Bryopsidales

The proposed link between anisogamy and adult size and complexity (PBS, [3]) depends on zygotic provisioning being translated eventually into the future success of the adult. In haplodiplontic ulvophycean green algal taxa, the zygote does not generate the haploid adult (gametophyte) directly, but instead, zygotic reserves translate into the diploid adult (sporophyte) as in diplontic taxa (cf. figure 1*b,c*). However, unlike diplontic taxa, haplodiplontic species may first undergo many mitotic divisions followed by the formation of meiotic progeny (zoospores) [19], each of which is released and germinates to produce a haploid adult (gametophyte) (figure 1*c*). In such species, there is, therefore, no direct translation of zygotic reserves into the haploid adult (gametophyte). Note that in Bryopsidales, we have shown that the anisogamy ratio is positively correlated with the complexity of the diploid adult (sporophyte) stage that is directly provisioned by the zygote (figure 3*b*; see also electronic supplementary material, figure S3*b*), but not to the final adult sporophyte size (figure 3*a*; see also electronic supplementary material, figure S3*a*). Increased size of the zygote might increase its survival at an early stage of development under severe environmental conditions. Morphological complexity of the diploid adult (sporophyte) may also be driven by increased environmental complexity.

That zygote provisioning seems unrelated to the final size of the diploid sporophyte stage at first appears puzzling: a typical example is the giant diploid stage of *Caulerpa racemosa* with a moderate anisogamy ratio of 1.9 (electronic supplementary material, tables S1 and S2; see also [48]). However, the higher survival rate of zygotes may not necessarily increase final diploid sporophyte stage size. Under severe environmental conditions, Bryopsidales may need complex systems to survive: for example, three-dimensional complex body structure would be useful for efficient photosynthesis in deep water. But it might be difficult to develop a large body under such conditions.

Can there be an indirect link between zygote size and haploid adult (gametophyte) size/complexity? For haploid adult (gametophyte) size and complexity to explain anisogamy evolution requires that increased diploid zygote size becomes more important in provisioning the haploid adult (gametophyte) as its size and complexity increases. While this might be possible in haplontic taxa such as volvocines (see §4.3. below), in many haplodiplontic ulvophyceans including some species of Bryopsidales (e.g. figure 1*c*), the zygote first undergoes considerable growth to form the sporophyte, before releasing numerous zoospores [19], so there is unlikely to be any direct provisioning link between the product of syngamy (the zygote) and the haploid adult gametophyte. Any such provisioning will be through the zoospore. In such cases, the size (and hence number) of the zoospores produced by the sporophyte is not likely to be regulated by the game-theoretical evolutionary process involved in GD theory for anisogamy, but by some marginal value optimization process of the form outlined by Smith & Fretwell [49].

## 4.3. Anisogamy in volvocine algae

Volvocine freshwater green algae have been the traditional taxon for testing the GD model [10,31,38,46,50,51], and the most important empirically testable prediction has been a positive correlation between the anisogamy ratio and haploid gametophytic adult colony size [6]. Colonial species release sperm packets and ova are fertilized internally, within nearby colonies [10,38,51], though this does not significantly alter GD predictions [6]. Some caution is necessary as to whether the GD model predicts that the anisogamy ratio should increase with colony size in volvocines, because each colony is not directly provisioned by the zygote [10]. Volvocines typically exhibit haplontic life cycles: the zygote does not grow into a diploid adult (sporophyte) as in Bryopsidales, but instead becomes a unicellular diploid zygospore with a thick cell wall, which divides meiotically into one, two or four gone cells (figure 1*a*). Each gone cell develops into a haploid adult colony

(gametophyte) that produces gametes mitotically [14]. The translation of zygotic resources into the production of the haploid gametophytic adult colony is, therefore, much more direct than in Bryopsidales, and some correlation between zygote size and gone cell size seems inevitable in volvocines, so that zygotic provisioning of the gametophyte via gone cell mass remains possible. Larger zygospores may produce larger meiotic progeny that then develop into larger colonies [10], compatible with GD expectation.

Indeed, since Knowlton [31], many authors have shown that anisogamy correlates with haploid gametophytic adult colonial size and complexity in volvocines, and Hanschen *et al.* [10] have recently supported the idea that multicellularity (body size and complexity) drove the evolution of volvocine anisogamy [10]. They showed that zygote size correlates positively with the anisogamy ratio in volvocines, and that the anisogamy ratio correlates positively with haploid gametophytic adult colony size and complexity, but found no relationship between a reduced number of meiotic products and haploid gametophytic adult colony size and complexity.

Additionally, the prediction that multicellularity drives anisogamy in green algae encounters anomalies [46]. Both isogamy and anisogamy are found in small unicellular species and large multicellular species within a wide size range (see fig. 2 in [52]). The retention of isogamy in large multicellular species has been explained as a consequence of a low gamete encounter rate: gametes are constrained to be large because well-provisioned gametes are needed to survive for extended periods before sexual fusion [52]. However, anisogamy in small unicellular species remains unexplained.

So what is 'adult size and complexity' that anisogamy evolves with? Caution is needed: this relationship demands a direct causal connection between zygote size and adult stage. Thus, while statistical analyses of comparative data appear to confirm that both anisogamy ratio and zygote size increase with the size of colonies in volvocine algae [10,38,46], to demonstrate that increasing adult size/complexity drives increasing zygote size through the increased need for resources ideally also requires positive correlations: (i) between zygospore mass and meiotic product (gone cell) size (plausible, especially in species with just one gone cell; figure 1*a*), and (ii) between gone cell size and haploid gametophytic adult colony size/complexity.

However, the fact that zygote size correlates with the transition to anisogamy in volvocines is consistent with GD theory, whether it reflects selection to increase zygote size alone (our hypothesis for ulvophycean algae), or selection to increase zygote size for provisioning the haploid adult colony (the classical theory) via increased gone cell size. It will be difficult to separate these two explanations for anisogamy (zygote size–complexity or adult size–complexity) in haplontic life cycles that have a close correlation between zygote size and meiotic product size (figure 1*a*), and in the many taxa with diplontic life cycles, i.e. animals and some algae (figure 1*b*). Our results for Bryopsidales suggest an alternative driver of anisogamy—differing demands of habitat environment, leading to zygote size variation. The environment might have played an important role in the evolution of anisogamy also in volvocines [50] though insufficient information on volvocine habitats is available to test this hypothesis [10]. So is the driver habitat or adult size/complexity, or some combination of the two?

# 5. Conclusion

Bryopsidales comparative data support the GD theory: selection for isogamy–anisogamy depends on an evolutionary game between the gametes of different mating types over investment in the zygote at syngamy. In haplodiplontic Bryopsidales, zygote investment transforms into zoospores, generated by many mitotic divisions and meiosis, which produce the haploid adults (gametophytes). For these taxa, we propose that it will be the complexity of the diploid adult sporophyte stage provisioned directly by the zygote, not the haploid adult (gametophyte), that selects for anisogamy, and that the haploid zoospore size will be determined by a different process (such as [49]). In diplontic Bryopsidales, zygotes generate the diploid adult sporophyte stage, so provisioning is direct. In both cases, it is the diploid (not haploid) stage complexity that appears important for GD theory.

Data accessibility. All data are available in the electronic supplementary material, tables S1 and S2.

Authors' contributions. T.T. designed the study. T.T. and Y.H. collected the data. T.T., Y.H. and G.A.P. analysed the data. T.T. and G.A.P. wrote the paper. All authors gave final approval of the publication.

Competing interests. The authors declare no competing interests.

Funding. This research was supported by grants-in-aid from the Japan Society for the Promotion of Science (no. 16H04839 to T.T.).

Acknowledgements. We thank Yutaro Sugii for his technical support.

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
