## [Peer Review File · Royal Society Open Science]

Review History

RSOS-201611.R0 (Original submission)

Review form: Reviewer 1

Is the manuscript scientifically sound in its present form?

Yes

Are the interpretations and conclusions justified by the results?

No

Is the language acceptable?

Yes

Do you have any ethical concerns with this paper?

No

Have you any concerns about statistical analyses in this paper?

No

Recommendation?

Major revision is needed (please make suggestions in comments)

Comments to the Author(s)

This paper argues for the dependence of adult size and complexity with the anisogamy ratio. But a simpler interpretation of Figure 2 would be the relationship between the female gamete size and them. This is because the variation of the size of male gametes is very small (Fig. S2), and only the size of female gametes differs between species. There is almost no difference between Fig. 3 with the anisogamy ratio on the vertical axis and Fig. S3 with zygote size.

According to the gamete dynamics (GD) theory, the size of male gametes should be considered to be minimized under the developmental constraints. The analysis in this paper does not say anything about why male gamete size variations are so small. This issue seems to be interesting. Another interesting point is that variation in female gamete size may be related to adult complexity, rather than discussing the anisogamy ratio. It should be necessary to discuss the relationship between and fitness. I think the interpretation of Figure 2 is rather misleading in GD theory. The author draws a scatter plot of the size variation between male and female gametes in Figure 2.

I would like a little more detailed analysis of the relationship between anisogamy ratio (or zygote size) and the adult complexity. There is not necessarily a relationship between closely related species that the bigger adult needs the larger zygote or egg as shown in Fig. 3(a). The survival rate of zygotes (eggs) may be an increasing function of their size, but it is unclear whether the number of eggs should be reduced and increased. This is a breakthrough in the PBS (Parker, Barker, Smith) model, but this article does not take advantage of it.

Relationship between size and fitness In GD theory, we should first discuss the relationship between the zygote size and the zygote fitness. This relationship is understandable if there is a relationship that the survival rate is low or zero unless the zygotes are large when the adult is complicated.

This paper certainly summarized the relationship between female gamete size and adult complexity. It also summarizes things that have nothing to do with adult size. It could be said that the authors presented such new findings, but I think they have just raised the issue but give no verification.

Review form: Reviewer 2

Is the manuscript scientifically sound in its present form?

Yes

Are the interpretations and conclusions justified by the results?

No

Is the language acceptable?

Yes

Do you have any ethical concerns with this paper?

No

Have you any concerns about statistical analyses in this paper?

No

Recommendation?

Major revision is needed (please make suggestions in comments)

Comments to the Author(s)

This is an interesting test of a theory of the evolution of anisogamy. It is particularly useful in dissecting the link between zygote size and adult size. However, there is no justification given for combining the original gamete competition model with what is called the gamete limitation model and calling the new model the gamete dynamics model. The gamete limitation model does not appear to have any support (da Silva and Drysdale 2018; Parker & Lehtonen 2014) and it is based on a model that erroneously uses absolute fitnesses when it should be using relative fitnesses within mating types (Lehtonen & Kokko 2011). All of the results support the gamete competition model on its own and no results are given that would require gamete limitation. Therefore, it would be more parsimonious to frame this work as a test of the gamete competition model, possibly with some discussion of gamete limitation, if necessary.

Also, on p. 6, lines 155-157, there is no justification for measuring diploid sporophyte volume as the cube of the maximum length. Is the sporophyte assumed to be a cube? Perhaps this explains the lack of correlations with this measure.

Review form: Reviewer 3

Is the manuscript scientifically sound in its present form?

Yes

Are the interpretations and conclusions justified by the results?

No

Is the language acceptable?

Yes

Do you have any ethical concerns with this paper?

No

Have you any concerns about statistical analyses in this paper?

No

Recommendation?

Accept with minor revision (please list in comments)

Comments to the Author(s)

In the manuscript "A comparative test of the gamete dynamics theory for the evolution of anisogamy in Bryopsidales green algae" Togashi, Horinouchi, and Parker investigate predictions of gamete dynamic theory using data from a group of algae that display interesting life cycles. Especially interesting is that many species are haplodiplontic with adult haploid and adult diploid stages. This life cycle allows the authors to test the production that female gametes are large to provide resources to the adult stage. I enjoyed reading the paper - it is quite clean in that it is concisely written and the predictions are clearly stated and the data are well analyzed to address the predictions. My main concern is that I am not totally convinced of some of the interpretations offered - this stems largely from the correlative nature of the dataset .

1. Stress - I am not totally convinced by the idea that species that live in 'stressful' environments are in fact stressed. Most organisms that live in such environments are adapted to the environment and would only be stressed by leaving their native environment. Additionally it's unclear to me why 'stress' of this kind would impact the evolutionary trade-off of resource provisioning of female gametes. The argument made is that the extra resources in gametes are more valuable in the stressful environment - but the opposite prediction would be that more, smaller female gametes might increase fitness by increasing the chances of creating more zygotes, when gametes might be lost to chance or go unfertilized. Is there theory and/or empirical data to support that the trade-off of size and number of female gametes is controlled by stress? Is there evidence that these algae are stressed?

2. Eyespot - An alternate explanation for the larger gametes in the stressful environment is that losing the eyespot relieves the energy and morphological requirement for those female gametes - which allows them to become bigger. Are these female gametes still motile? Does motility impose shape/size restrictions? Additionally, if the lack of phototactic behaviour may make it harder for male and female gametes to 'find' one another; could the larger size be associated with the longevity of the female gamete in the environment?

3. The correlation of complexity and anisogamy looks to be driven by two data points - this isn't really alluded to in the paper - given that phylogeny is accounted for in this analysis I assume these species aren't closely related. Can you expand on these two data points?

Minor comments:

The abstract would be slightly better if the nature of the data were better explained - most of the abstract is background and the data analyzed are mostly left out which makes the results/interpretations hard to understand until after one has read the paper.

Figure 1 - many readers will have to put some effort in to understand these life cycles and the associated terminology - I think this figure is key and would be more clear if you labeled the panels with haplontic/diplontic/haplodiplontic

Decision letter (RSOS-201611.R0)

Dear Dr Togashi

The Editors assigned to your paper RSOS-201611 "A comparative test of the gamete dynamics theory for the evolution of anisogamy in Bryopsidales green algae" have now received comments from reviewers and would like you to revise the paper in accordance with the reviewer comments and any comments from the Editors. Please note this decision does not guarantee eventual acceptance.

We do not generally allow multiple rounds of revision so we urge you to make every effort to fully address all of the comments at this stage. If deemed necessary by the Editors, your

manuscript will be sent back to one or more of the original reviewers for assessment. If the original reviewers are not available, we may invite new reviewers.

Please submit your revised manuscript and required files (see below) no later than 21 days from today's (ie 18-Nov-2020) date. Note: the ScholarOne system will 'lock' if submission of the revision is attempted 21 or more days after the deadline. If you do not think you will be able to meet this deadline please contact the editorial office immediately.

on behalf of Dr Punidan Jeyasingh (Associate Editor) and Pete Smith (Subject Editor)
openscience@royalsociety.org

Associate Editor Comments to Author (Dr Punidan Jeyasingh):

Associate Editor: 1

Comments to the Author:

I was very excited about this paper during pre-assessment. It maps ecology onto well developed theory to explain a longstanding evolutionary issue. The subject area is quite specific, and I had a hard time finding suitable reviewers. I am most grateful to the three experts for agreeing to review the manuscript. While all three experts were enthusiastic about the work, they have also raised a number of issues. I felt the comments were fair, clear, and constructive. With much gratitude to the experts, I invite the authors to incorporate these comments and submit a revised version of the manuscript.

Reviewer comments to Author:

Reviewer: 1

Comments to the Author(s)

This paper argues for the dependence of adult size and complexity with the anisogamy ratio. But a simpler interpretation of Figure 2 would be the relationship between the female gamete size and them. This is because the variation of the size of male gametes is very small (Fig. S2), and only the size of female gametes differs between species. There is almost no difference between Fig. 3 with the anisogamy ratio on the vertical axis and Fig. S3 with zygote size.

According to the gamete dynamics (GD) theory, the size of male gametes should be considered to be minimized under the developmental constraints. The analysis in this paper does not say anything about why male gamete size variations are so small. This issue seems to be interesting. Another interesting point is that variation in female gamete size may be related to adult complexity, rather than discussing the anisogamy ratio. It should be necessary to discuss the

relationship between and fitness. I think the interpretation of Figure 2 is rather misleading in GD theory. The author draws a scatter plot of the size variation between male and female gametes in Figure 2.

I would like a little more detailed analysis of the relationship between anisogamy ratio (or zygote size) and the adult complexity. There is not necessarily a relationship between closely related species that the bigger adult needs the larger zygote or egg as shown in Fig. 3(a). The survival rate of zygotes (eggs) may be an increasing function of their size, but it is unclear whether the number of eggs should be reduced and increased. This is a breakthrough in the PBS (Parker, Barker, Smith) model, but this article does not take advantage of it.

Relationship between size and fitness In GD theory, we should first discuss the relationship between the zygote size and the zygote fitness. This relationship is understandable if there is a relationship that the survival rate is low or zero unless the zygotes are large when the adult is complicated.

This paper certainly summarized the relationship between female gamete size and adult complexity. It also summarizes things that have nothing to do with adult size. It could be said that the authors presented such new findings, but I think they have just raised the issue but give no verification.

Reviewer: 2

Comments to the Author(s)

This is an interesting test of a theory of the evolution of anisogamy. It is particularly useful in dissecting the link between zygote size and adult size. However, there is no justification given for combining the original gamete competition model with what is called the gamete limitation model and calling the new model the gamete dynamics model. The gamete limitation model does not appear to have any support (da Silva and Drysdale 2018; Parker & Lehtonen 2014) and it is based on a model that erroneously uses absolute fitnesses when it should be using relative fitnesses within mating types (Lehtonen & Kokko 2011). All of the results support the gamete competition model on its own and no results are given that would require gamete limitation. Therefore, it would be more parsimonious to frame this work as a test of the gamete competition model, possibly with some discussion of gamete limitation, if necessary.

Also, on p. 6, lines 155-157, there is no justification for measuring diploid sporophyte volume as the cube of the maximum length. Is the sporophyte assumed to be a cube? Perhaps this explains the lack of correlations with this measure.

Reviewer: 3

Comments to the Author(s)

In the manuscript "A comparative test of the gamete dynamics theory for the evolution of anisogamy in Bryopsidales green algae" Togashi, Horinouchi, and Parker investigate predictions of gamete dynamic theory using data from a group of algae that display interesting life cycles. Especially interesting is that many species are haplodiplontic with adult haploid and adult diploid stages. This life cycle allows the authors to test the production that female gametes are large to provide resources to the adult stage. I enjoyed reading the paper - it is quite clean in that it is concisely written and the predictions are clearly stated and the data are well analyzed to address the predictions. My main concern is that I am not totally convinced of some of the interpretations offered - this stems largely from the correlative nature of the dataset .

1. Stress - I am not totally convinced by the idea that species that live in 'stressful' environments are in fact stressed. Most organisms that live in such environments are adapted to the environment and would only be stressed by leaving their native environment. Additionally it's unclear to me why 'stress' of this kind would impact the evolutionary trade-off of resource provisioning of female gametes. The argument made is that the extra resources in gametes are more valuable in the stressful environment - but the opposite prediction would be that more, smaller female gametes might increase fitness by increasing the chances of creating more zygotes, when gametes might be lost to chance or go unfertilized. Is there theory and/or empirical data to support that the trade-off of size and number of female gametes is controlled by stress? Is there evidence that these algae are stressed?

2. Eyespot - An alternate explanation for the larger gametes in the stressful environment is that losing the eyespot relieves the energy and morphological requirement for those female gametes - which allows them to become bigger. Are these female gametes still motile? Does motility impose shape/size restrictions? Additionally, if the lack of phototactic behaviour may make it harder for male and female gametes to 'find' one another; could the larger size be associated with the longevity of the female gamete in the environment?

3. The correlation of complexity and anisogamy looks to be driven by two data points - this isn't really alluded to in the paper - given that phylogeny is accounted for in this analysis I assume these species aren't closely related. Can you expand on these two data points?

Minor comments:

The abstract would be slightly better if the nature of the data were better explained - most of the abstract is background and the data analyzed are mostly left out which makes the results/interpretations hard to understand until after one has read the paper.

Figure 1 - many readers will have to put some effort in to understand these life cycles and the associated terminology - I think this figure is key and would be more clear if you labeled the panels with haplontic/diplontic/haplodiplontic

===PREPARING YOUR MANUSCRIPT===

===PREPARING YOUR REVISION IN SCHOLARONE===

Author's Response to Decision Letter for (RSOS-201611.R0)

See Appendix A.

RSOS-201611.R1 (Revision)

Review form: Reviewer 1

Is the manuscript scientifically sound in its present form?

Yes

Are the interpretations and conclusions justified by the results?

No

Is the language acceptable?

Yes

Do you have any ethical concerns with this paper?

No

Have you any concerns about statistical analyses in this paper?

No

Recommendation?

Accept with minor revision (please list in comments)

Comments to the Author(s)

I think this paper states the following three points. If so, these should be stated clearly: (1) Among the algae examined in this article, the variation in the size of male gametes is extremely small, and it is considered that the male gamete sizes are minimized. (2) It is considered that the size of female gametes is optimized by the tradeoff between the survival rate and the number of female gametes in the zygote stage, not in adults. (3) Since the gamete size and the adult size do not correlate, it does not necessarily optimize the relationship between the size of the adult and the fitness in adult stage

Review form: Reviewer 2

Is the manuscript scientifically sound in its present form?

No

Are the interpretations and conclusions justified by the results?

No

Is the language acceptable?

Yes

Do you have any ethical concerns with this paper?

No

Have you any concerns about statistical analyses in this paper?

No

Recommendation?

Reject

Comments to the Author(s)

None of my comments have been adequately addressed.

Decision letter (RSOS-201611.R1)

Dear Dr Togashi

The Editors assigned to your paper RSOS-201611.R1 "A comparative test of the gamete dynamics theory for the evolution of anisogamy in Bryopsidales green algae" have now received comments from reviewers and would like you to revise the paper in accordance with the reviewer comments and any comments from the Editors. Please note this decision does not guarantee eventual acceptance.

We do not generally allow multiple rounds of revision so we urge you to make every effort to fully address all of the comments at this stage, and it is rare for the Editors to invite a further round of major revision as they have done here -- there will be no further opportunities to revise your paper, so please do make sure you fully address all of the concerns raised by the Editors and the reviewers in your next iteration. If deemed necessary by the Editors, your manuscript will be sent back to one or more of the original reviewers for assessment. If the original reviewers are not available, we may invite new reviewers.

Please submit your revised manuscript and required files (see below) no later than 21 days from today's (ie 13-Jan-2021) date. Note: the ScholarOne system will 'lock' if submission of the revision is attempted 21 or more days after the deadline. If you do not think you will be able to meet this deadline please contact the editorial office immediately.

on behalf of Dr Punidan Jeyasingh (Associate Editor) and Pete Smith (Subject Editor)
openscience@royalsociety.org

Associate Editor Comments to Author (Dr Punidan Jeyasingh):

Associate Editor: 1

Comments to the Author:

I thank the authors for submitting a revised version of the manuscript. This version was reassessed by two experts. While one reviewer was generally happy with the revisions (although some issues remain), another was clearly not happy with how their comments were addressed (although they have not provided specifics). I agree with reviewer 1. The manuscript can be made clearer. I too am a bit unclear on the main points the manuscript wants readers to take home. I invite the authors to do a mid-major revision of the manuscript particularly keeping the message and terms consistent throughout.

Reviewer comments to Author:

Reviewer: 1

Comments to the Author(s)

I think this paper states the following three points. If so, these should be stated clearly: (1) Among the algae examined in this article, the variation in the size of male gametes is extremely small, and it is considered that the male gamete sizes are minimized. (2) It is considered that the size of female gametes is optimized by the tradeoff between the survival rate and the number of female gametes in the zygote stage, not in adults. (3) Since the gamete size and the adult size do not correlate, it does not necessarily optimize the relationship between the size of the adult and the fitness in adult stage

Reviewer: 2

Comments to the Author(s)

None of my comments have been adequately addressed.

===PREPARING YOUR MANUSCRIPT===

===PREPARING YOUR REVISION IN SCHOLARONE===

-- If you have uploaded ESM files, please ensure you follow the guidance at <https://royalsociety.org/journals/authors/author-guidelines/#supplementary-material> to include a suitable title and informative caption. An example of appropriate titling and captioning may be found at https://figshare.com/articles/Table_S2_from_Is_there_a_trade-off_between_peak_performance_and_performance_breadth_across_temperatures_for_aerobic_sc_ope_in_teleost_fishes_/3843624.

Author's Response to Decision Letter for (RSOS-201611.R1)

See Appendix B.

Decision letter (RSOS-201611.R2)

Dear Dr Togashi,

It is a pleasure to accept your manuscript entitled "A comparative test of the gamete dynamics theory for the evolution of anisogamy in Bryopsidales green algae" in its current form for publication in Royal Society Open Science.

You can expect to receive a proof of your article in the near future. Please contact the editorial office (openscience@royalsociety.org) and the production office (openscience_proofs@royalsociety.org) to let us know if you are likely to be away from e-mail contact – if you are going to be away, please nominate a co-author (if available) to manage the proofing process, and ensure they are copied into your email to the journal.

on behalf of Dr Punidan Jeyasingh (Associate Editor) and Pete Smith (Subject Editor)
openscience@royalsociety.org

Associate Editor Comments to Author (Dr Punidan Jeyasingh):
Associate Editor
Comments to the Author:

I thank the authors for making these final tweaks to make the manuscript more accessible. I am grateful to the expert reviewers for constructive comments, and to the authors for addressing them. I am happy to recommend this manuscript for publication.

Appendix A

Dr Punidan Jeyasingh
Royal Society Open Science
December 2 2020

Dear Dr Jeyasingh:

Subject: Revision of manuscript RSOS-201611.

Thank you for your email dated 18 Nov 2020 with the reviewers' and your comments enclosed. We have carefully addressed all comments in depth and revised the manuscript accordingly. Our responses are given in a point-by-point manner below and include the corresponding changes made to our manuscript. Changes to the manuscript are highlighted in red.

We hope the revised version is now suitable for publication.

Very truly yours,

Tatsuya Togashi Ph.D.
Marine Biosystems Research Center, Chiba University, Japan

Associate Editor Comments to Author (Dr Punidan Jeyasingh):

Associate Editor: 1

Comments to the Author:

I was very excited about this paper during pre-assessment. It maps ecology onto well developed theory to explain a longstanding evolutionary issue. The subject area is quite specific, and I had a hard time finding suitable reviewers. I am most grateful to the three experts for agreeing to review the manuscript. While all three experts were enthusiastic about the work, they have also raised a number of issues. I felt the comments were fair, clear, and constructive. With much gratitude to the experts, I invite the authors to incorporate these comments and submit a revised version of the manuscript.

REPLY 1: We much appreciate the time and effort spent by the three reviewers, and thank them sincerely for their comments and suggestions, which have been very helpful in revising and improving the manuscript.

Reviewer comments to Author:

Reviewer: 1

Comments to the Author(s)

This paper argues for the dependence of adult size and complexity with the anisogamy ratio. But a simpler interpretation of Figure 2 would be the relationship between the female gamete size and them. This is because the variation of the size of male gametes is very small (Fig. S2), and only the size of female gametes differs between species. There is almost no difference between Fig. 3 with the anisogamy ratio on the vertical axis and Fig. S3 with zygote size.

REPLY 2: Yes, this is true. But the ovum provisioning level will be mostly optimized by the needs of the zygote. We believe that because of wide variation in environmental constraints, this will generate considerable variation in the size of the zygote.

According to the gamete dynamics (GD) theory, the size of male gametes should be considered to be minimized under the developmental constraints. The analysis in this paper does not say anything about why male gamete size variations are so small. This issue seems to be interesting.

REPLY 3: In contrast to the female gamete and hence the zygote, the male gamete has primarily a transient adaptive function – that of fusion. We believe that this transient (and relatively invariant) function is the main reason why male gamete minimization has occurred, in conformity with GD theory. We have added the following:

Page 10 Line 255-259

In contrast to the female gamete and hence the zygote, the male gamete has primarily a transient adaptive function – that of fusion. We suggest that this transient (and relatively invariant) function is the main reason why male gamete minimization has occurred, in conformity with GD theory.

Another interesting point is that variation in female gamete size may be related to adult complexity, rather than discussing the anisogamy ratio. It should be necessary to discuss the relationship between and fitness. I think the interpretation of Figure 2 is rather misleading in GD theory. The author draws a scatter plot of the size variation between male and female gametes in Figure 2.

REPLY 4: In Fig. 2, we have attempted to outline why we believe that there is high

variation in anisogamy ratio – because of high variation in environmental stress levels generated by habitat, in terms of sea depth. We have argued that lower light and other resource levels create a demand for more resources (e.g. more chloroplasts). Fig. 2(a) shows that male gamete size is minimized (they are transient and selected predominantly for fusion), while female gametes vary in size due to the demands of the zygote.

I would like a little more detailed analysis of the relationship between anisogamy ratio (or zygote size) and the adult complexity. There is not necessarily a relationship between closely related species that the bigger adult needs the larger zygote or egg as shown in Fig. 3(a). The survival rate of zygotes (eggs) may be an increasing function of their size, but it is unclear whether the number of eggs should be reduced and increased. This is a breakthrough in the PBS (Parker, Barker, Smith) model, but this article does not take advantage of it.

REPLY 5: The reviewer has a fair point here – we thought initially that we would find that the anisogamy ratio increases with diploid sporophyte size (mass), as we have discussed during the preparation of the paper. However, as we described in the discussion section, in ulvophyceans it is difficult probably to develop a large body under colder, light-limited environments (deep waters). We could not directly test whether anisogamy ratio increases with sea depth, but we provide empirical arguments why complexity should increase, in terms of eyespot and phototactic behaviour of gametes. We anticipate that the correlation between anisogamy ratio and diploid stage complexity is due to a correlation between the need to increase ovum size (to increase zygote survival) and to increased diploid stage complexity, both to cope with the severe conditions associated with increased sea depth.

Concerning gamete size, as described in the introduction section, a size-number trade off exists in female gametes (eggs). Also, to clarify our finding, we added a sentence to the discussion section.

Page 3 Line 47-50

(ii) a size-number trade off applies, i.e. gamete size decreases in inverse proportion to the number of gametes produced from a given total gametic investment, a trait confirmed empirically in many organisms including algae (e.g. [11]).

Page 11 Line 301-302

The higher survival rate of zygotes may not necessarily increase final diploid stage size.

Relationship between size and fitness In GD theory, we should first discuss the relationship between the zygote size and the zygote fitness. This relationship is understandable if there is a relationship that the survival rate is low or zero unless the zygotes are large when the adult is complicated.

REPLY 6: Thank you very much for this interesting idea. We have discussed the relationship between final-size diploid stage complexity and its fitness. We will provide direct empirical data on the relationship between zygote size and its fitness in a future paper.

Page 11 Line 302-306

Under **severe environmental** conditions, Bryopsidales may need complex systems to survive: for example, three-dimensional complex body structure would be useful for efficient photosynthesis in deep water. But it might be difficult to develop a large body under such conditions.

This paper certainly summarized the relationship between female gamete size and adult complexity. It also summarizes things that have nothing to do with adult size. It could be said that the authors presented such new findings, but I think they have just raised the issue but give no verification.

REPLY 7: Our prediction was that in diplontic species, ovum (hence zygote) mass is expected to correlate with adult mass and complexity, directly following GD theory. However, in haplodiplontic species we do not expect the zygote mass to correlate with adult mass and complexity: our point is that in such species it should relate to mass and complexity of the final-size diploid sporophyte stage, since this typically releases many small zoospores that later grow into the gametophytic adult. This is an important modification of GD theory. We find that across Bryopsidales, zygote mass is indeed related to complexity of the diploid stage in diplontic and haplodiplontic species, as we predict, though we were unable to find the expected relationship with mass of them. We have attempted to explain why this is so in the discussion section (see REPLY 5).

Reviewer: 2

Comments to the Author(s)

This is an interesting test of a theory of the evolution of anisogamy. It is particularly useful in dissecting the link between zygote size and adult size. However, there is no justification given for combining the original gamete competition model with what is called the gamete limitation model and calling the new model the gamete dynamics model. The gamete limitation model does not appear to have any support (da Silva and Drysdale 2018; Parker & Lehtonen 2014) and it is based on a model that erroneously uses absolute fitnesses when it should be using relative fitnesses within mating types (Lehtonen & Kokko 2011). All of the results support the gamete competition model on its own and no results are given that would require gamete limitation. Therefore, it would be more parsimonious to frame this work as a test of the gamete competition model, possibly with some discussion of gamete limitation, if necessary.

REPLY 8: We fully agree with the reviewer that gamete competition is generally likely to form the strongest reason for the evolution of anisogamy both theoretically and empirically. Since the two models (gamete competition and gamete limitation) are identical in all other assumptions, it seemed sensible to combine them (as did Lehtonen & Kokko, 2011) so that both factors could vary continuously, as they do in broadcast spawning taxa in nature, thus forming a more complete biological model that is applicable across the full range of natural situations. We do agree, however, with the reviewer that ulvophyceans with phototactic gametes and/or sexual pheromones might not be often under gamete limiting conditions and have stressed that gamete competition is more likely to account for our results.

Page 10 Line 266-268

Note that ulvophyceans with phototactic gametes and/or sexual pheromones might not be often under gamete limiting conditions, and that gamete competition is more likely to account for our results.

We were interested in the reviewer's criticism of the model of Lehtonen & Kokko (2011). The results presented in their paper are evolutionary trajectories and equilibria. The direction of a trajectory at any point in the figures in their paper appears to be determined by the relative magnitudes of selection on the two mating types. In their model, the application of the 'Fisher condition' implies that mean fitness of the two mating types must be identical at every point in the graphs. Therefore, if absolute fitness was divided by mean fitness to get relative fitness, the divisor would be identical for both mating types. Hence relative magnitudes of selection on the mating types would not be affected and the directions in the trajectories would not be affected either.

Although we fully agree that it can be important to distinguish between absolute and relative fitness in some models, we do not think that this distinction affects the results in Lehtonen & Kokko's model, and so do not believe that their results are erroneous.

Also, on p. 6, lines 155-157, there is no justification for measuring diploid sporophyte volume as the cube of the maximum length. Is the sporophyte assumed to be a cube? Perhaps this explains the lack of correlations with this measure.

REPLY 9: Thank you for this comment. In Bryopsidales, diploid sporophytes are generally filamentous. So we calculated the relative volume of diploid sporophytes.

Page 6 Line 155-159

We took the cube of the maximum length of the **final-size** sporophytes to represent the **relative** volumetric size of adult diploid sporophytes in each species, **since Bryopsidales sporophytes are often filamentous, and their diameter appears to increase in roughly constant proportion to their maximum length, so that maximum length cubed gives a measure of relative volume.**

Reviewer: 3

Comments to the Author(s)

In the manuscript "A comparative test of the gamete dynamics theory for the evolution of anisogamy in Bryopsidales green algae" Togashi, Horinouchi, and Parker investigate predictions of gamete dynamic theory using data from a group of algae that display interesting life cycles. Especially interesting is that many species are haplodiplontic with adult haploid and adult diploid stages. This life cycle allows the authors to test the production that female gametes are large to provide resources to the adult stage. I enjoyed reading the paper - it is quite clean in that it is concisely written and the predictions are clearly stated and the data are well analyzed to address the predictions. My main concern is that I am not totally convinced of some of the interpretations offered - this stems largely from the correlative nature of the dataset .

1. Stress - I am not totally convinced by the idea that species that live in 'stressful' environments are in fact stressed. Most organisms that live in such environments are adapted to the environment and would only be stressed by leaving their native

environment. Additionally it's unclear to me why 'stress' of this kind would impact the evolutionary trade-off of resource provisioning of female gametes. The argument made is that the extra resources in gametes are more valuable in the stressful environment - but the opposite prediction would be that more, smaller female gametes might increase fitness by increasing the chances of creating more zygotes, when gametes might be lost to chance or go unfertilized. Is there theory and/or empirical data to support that the trade-off of size and number of female gametes is controlled by stress? Is there evidence that these algae are stressed?

REPLY 10: We thank the reviewer for this comment. Perhaps our use of 'stressful' was inappropriate, we now use 'environmental severity' throughout the manuscript not only in the abstract. Ulvophyceans including Bryopsidales green algae appear to increase zygote size adapting to deep waters. In deep waters, large zygotes would be advantageous for survival and hence safe development adapting to weak light and cold water. In other words, most of small zygotes would not be able to survive in such environments though more zygotes might be produced. Our previous theoretical study supports that the trade-off of size and number of female gametes is controlled by environmental severity (Togashi et al. 2012). Additionally, gamete size and its phototactic behaviour empirically support it. We added a phrase to the discussion section.

Page 5 Line 107

severity

Page 5 Line 129

restriction in

Page 10 Line 273

severe

Page 10 Line 278-280

since most of small zygotes would not be able to survive there even if more zygotes were produced

Page 11 Line 302-303

severe environmental

2. Eyespot - An alternate explanation for the larger gametes in the stressful environment is that losing the eyespot relieves the energy and morphological requirement for those female gametes - which allows them to become bigger. Are these female gametes still motile? Does motility impose shape/size restrictions? Additionally, if the lack of

phototactic behaviour may make it harder for male and female gametes to 'find' one another; could the larger size be associated with the longevity of the female gamete in the environment?

REPLY 11: This is an interesting comment. Thank you. First, we agree that losing the eyespot may slightly relieve the energy and morphological requirement for those female gametes. We added a sentence to the discussion section. Second, even in a species of Bryopsidales that produces female gametes with an eyespot, because of spawning synchrony all male gametes have died before female gametes begin to die (Togashi et al. 1998). Therefore, although eyespot loss might slightly increase the longevity of the female gametes, it would not be useful to produce more zygotes.

Page 10 Line 276-277

Losing the eyespot may permit energy to be diverted into chloroplasts.

3. 3. The correlation of complexity and anisogamy looks to be driven by two data points - this isn't really alluded to in the paper - given that phylogeny is accounted for in this analysis I assume these species aren't closely related. Can you expand on these data two points?

REPLY 12: Thank you for this question. We used PGLS (phylogenetic generalized least squares) incorporating phylogenetic information in this analysis and tested whether there is a relationship between two variables while accounting for the fact that lineages are not independent. So our result is free from this kind of problems.

Minor comments:

The abstract would be slightly better if the nature of the data were better explained - most of the abstract is background and the data analyzed are mostly left out which makes the results/interpretations hard to understand until after one has read the paper.

REPLY 13: Thank you very much for this comment. We have now explained more about how we derived our results/interpretations in the abstract.

Page 2 Line 14-30

Gamete dynamics theory proposes that anisogamy arises by disruptive selection for gamete numbers versus gamete size, and predicts that female/male gamete size (anisogamy ratio) will increase with adult size and complexity. Important evidence has been that in volvocine green algae, anisogamy ratio correlates positively with haploid

colony size. However, in green algae there are notable exceptions. We focus on Bryopsidales green algae. While some taxa have a diplontic life cycle in which a diploid adult arises directly from the zygote, many taxa have a haplodiplontic life cycle in which adults develop indirectly: the zygote first develops into a diploid sporophyte which later undergoes meiosis and releases (often many) haploid propagules, each growing into a haploid adult gametophyte. Using data from the literature, our comparative analyses suggest that minimized male gametes and female gametes of various sizes evolved as theory predicts, and that diploid (but not haploid) stage complexity correlates positively with anisogamy ratio. However, there was no correlation between final diploid stage mass and anisogamy ratio. Increased environmental severity (water depth) appears to drive increased diploid stage complexity and anisogamy ratio: gamete dynamics theory correctly predicts that anisogamy evolves with the (diploid) stage directly provisioned by the zygote.

Figure 1 - many readers will have to put some effort in to understand these life cycles and the associated terminology - I think this figure is key and would be more clear if you labeled the panels with haplontic/diplontic/haplodiplontic

REPLY 14: Thank you. We labelled the panels with haplontic/diplontic/haplodiplontic in figure 1.

END

Appendix B

Dr Punidan Jeyasingh
Royal Society Open Science
January 31 2021

Dear Dr Jeyasingh:

Subject: Revision of manuscript RSOS-201611.R1.

Thank you for your email dated 13 Jan 2021 with the reviewers' and your comments enclosed. We have carefully addressed all comments and revised the manuscript accordingly. Our responses are given below. Changes to the manuscript are highlighted in red.

We hope the revised version is now suitable for publication.

Very truly yours,

Tatsuya Togashi Ph.D.
Marine Biosystems Research Center, Chiba University, Japan

Associate Editor Comments to Author (Dr Punidan Jeyasingh):

Associate Editor: 1

Comments to the Author:

I thank the authors for submitting a revised version of the manuscript. This version was reassessed by two experts. While one reviewer was generally happy with the revisions (although some issues remain), another was clearly not happy with how their comments were addressed (although they have not provided specifics). I agree with reviewer 1. The manuscript can be made clearer. I too am a bit unclear on the main points the manuscript wants readers to take home. I invite the authors to do a mid-major revision of the manuscript particularly keeping the message and terms consistent throughout.

REPLY 1: Thank you for this comment. We have now standardised the terms throughout the manuscript, and also clarified our main points, as stressed by reviewer 1 (below): male gametes are minimized and female gametes have various sizes supporting gamete dynamics theory, and increased environmental severity (water depth) appears to drive the evolution of anisogamy suggesting that the gamete dynamics theory correctly predicts that anisogamy evolved with the (diploid) stage directly provisioned by the zygote.

Reviewer comments to Author:

Reviewer: 1

Comments to the Author(s)

I think this paper states the following three points. If so, these should be stated clearly: (1) Among the algae examined in this article, the variation in the size of male gametes is extremely small, and it is considered that the male gamete sizes are minimized. (2) It is considered that the size of female gametes is optimized by the tradeoff between the survival rate and the number of female gametes in the zygote stage, not in adults. (3) Since the gamete size and the adult size do not correlate, it does not necessarily optimize the relationship between the size of the adult and the fitness in adult stage

REPLY 2 See REPLY 1. Thank you very much again. We have attempted to stress these three main points, both in the Abstract and in the revised manuscript.

END